# Spin–orbit coupled molecular quantum magnetism realized in inorganic solid

Sang-Youn Park[1], S.-H. Do[1,2], K.-Y. Choi[2], J.-H. Kang[3], Dongjin Jang[3], B. Schmidt[3], Manuel Brando[3], B.-H. Kim[4], D.-H. Kim[1,5], N.P. Butch[6], Seongsu Lee[7], J.-H. Park[1,5,8] & Sungdae Ji[1,5]

Molecular quantum magnetism involving an isolated spin state is of particular interest due to the characteristic quantum phenomena underlying spin qubits or molecular spintronics for quantum information devices, as demonstrated in magnetic metal–organic molecular systems, the so-called molecular magnets. Here we report the molecular quantum magnetism realized in an inorganic solid $Ba_3Yb_2Zn_5O_{11}$ with spin–orbit coupled pseudospin-½ $Yb^{3+}$ ions. The magnetization represents the magnetic quantum values of an isolated $Yb_4$ tetrahedron with a total (pseudo)spin 0, 1 and 2. Inelastic neutron scattering results reveal that a large Dzyaloshinsky–Moriya interaction originating from strong spin–orbit coupling of Yb $4f$ is a key ingredient to explain magnetic excitations of the molecular magnet states. The Dzyaloshinsky–Moriya interaction allows a non-adiabatic quantum transition between avoided crossing energy levels, and also results in unexpected magnetic behaviours in conventional molecular magnets.

[1] Max Planck POSTECH Center for Complex Phase Materials, Pohang University of Science and Technology, Pohang 37673, Korea. [2] Department of Physics, Chung-Ang University, Seoul 06911, Korea. [3] Max Planck Institute for Chemical Physics in Solid, 01187 Dresden, Germany. [4] iTHES Research Group and Computational Condensed Matter Physics Laboratory, RIKEN, Wako, Saitama 351-0198, Japan. [5] Department of Physics, Pohang University of Science and Technology, Pohang 37673, Korea. [6] NIST Center for Neutron Research, National Institute of Standards and Technology, Gaithersburg, Maryland 20899, USA. [7] Neutron Science Division, HANARO, Korea Atomic Energy Research Institute, Daejeon 34057, Korea. [8] Division of Advanced Materials Science, Pohang University of Science and Technology, Pohang 37673, Korea. Correspondence and requests for materials should be addressed to J.-H.P. (email: jhp@postech.ac.kr) or to S.J. (email: sungdae@postech.ac.kr).

Quantum magnetism is a ubiquitous subject from the spin singlet state in non-interacting dimers to the long-range entangled state in quantum spin liquids[1–8]. Its nature is often described in terms of quantum values of the magnetic moments, ionic anisotropies and their coupling network[9–13]. The former two are determined by the single ionic characters, while the last one can be controlled by arrangement of the magnetic ions. Great progress in theoretical and experimental investigations on the quantum magnetism has mostly been made through variation of the coupling connectivity[3–13]. An extreme case could be the molecular magnet of an isolated magnetic cluster[14,15]. Meanwhile, magnetism in solids is strongly influenced by geometrical constraints such as frustration or bond alternation as well as dimensionality, and often introduces intriguing quantum phenomena[1,5–8,16].

With strong spin–orbit coupling (SOC), the magnetic quantum spin is given by the total angular momentum $J$, rather than the spin $S$. The degenerate $J$-state is split by a crystal field (CF), and the ground-state quantum spin can be represented by a modified $J$, the so-called pseudospin, as discussed in lanthanide-based molecular magnets[17–22]. The strong $4f$ SOC combined with a subtle CF contributes strong magnetic anisotropy to yield a simple model Hamiltonian with Ising-like or XY-like anisotropic magnetic exchange between the molecular pseudospins. Recently, $Ba_3Yb_2Zn_5O_{11}$ was reported to be a geometrically frustrated breathing pyrochlore system with two distinct Yb–Yb distances[23,24]. $Ba_3Yb_2Zn_5O_{11}$ consists of two alternating main blocks of $Yb_4O_{16}$ and $Zn_{10}O_{20}$ (Fig. 1a). The magnetic $Yb^{3+}$ ions in $Yb_4O_{16}$ form a tetrahedron connected with another one through corner-sharing in the three-dimensional framework. Remarkably, the inter-tetrahedron Yb–Yb distance $r' = 6.23$ Å is much larger than the intra-tetrahedron one $r = 3.30$ Å. Thus, the inter-magnetic exchange energy $J'$ becomes negligible in comparison with the intra-exchange energy $J$, that is, $J'/J \sim 0$ (Fig. 1b). As can be seen in the detailed ionic arrangements of the tetrahedron $Yb_4O_{16}$ (Fig. 1c), each Yb ion is surrounded by six oxygens to form an octahedron ($YbO_6$) with a trigonal distortion ($C_{3v}$ symmetry), and the $C_{3v}$ symmetry axis is towards the tetrahedron center. The $Yb^{3+}$ ion ($4f^{13}$) effectively has a pseudospin-½ ground state of a Kramers doublet separated by the CF splitting energy of 38.2 meV ($= 443$ K-$k_B$) without any magnetic long-range order even below sub-Kelvin in spite of the considerable Curie–Weiss temperature $\Theta_{CW} = -6.7$ K (refs 24,25), indicating possible formation of decoupled molecular spin states.

In this paper, we report the molecular magnetic behaviours in magnetization and inelastic neutron scattering (INS) for inorganic polycrystalline $Ba_3Yb_2Zn_5O_{11}$ samples. The magnetization shows the hysteretic step-like jumps between $S_{eff} = 0$, 1 and 2 molecular magnetic states of an isolated $Yb_4$ tetrahedron with spin–orbit coupled pseudospin-½, which reflects the non-adiabatic Landau–Zener transition. The INS measurement with external magnetic field unveils that the large Dzyaloshinsky–Moriya (DM) interaction originating from strong SOC of Yb $4f$ electron is essential to construct the $S_{eff}$ molecular magnetic states involving avoided level crossing. Our finding not only opens a possibility for qubit quantum device applications due to benefit of the regularity in the inorganic solid, but the tunable inter-tetrahedron distance also provides a playground to explore the crossover from isolated to entangled magnetic quantum systems in the presence of SOC.

## Results

**Magnetization and effective Hamiltonian.** Figure 1d shows the field-dependent magnetization result ($M$ versus $H$) at 100 mK,

which displays the characteristic step-like jumps of an anti-ferromagnetic (AFM) coupled molecule with a total (pseudo)spin $S_{eff}$ formed by the four ½-pseudospins of the tetrahedron. The ground state of the AFM-coupled tetrahedron, which is the $S_{eff} = 0$ state at zero field, is consecutively switched to the $S_{eff} = 1$ and the $S_{eff} = 2$ state as the external magnetic field increases across corresponding critical fields. The three plateaus in $M(H)$ represent the respective $S_{eff} = 0$, 1 and 2 states, and the consecutive level crossing quantum transitions are presented by the two-step feature at the critical fields $H_{C1} = 3.5$ T and $H_{C2} = 8.8$ T. It is worth to note that $M(H)$ exhibits three distinctive features: a hysteretic behaviour near $H_{C1}$, shifts of the critical fields with respect to the Heisenberg model and a non-zero slope in a lower field region $0 < H < H_{C1}$, all of which are not expected in conventional molecular magnets with a simple Heisenberg AFM exchange interaction. For comparison, the simulated $M(H)$ of the conventional molecular magnet with AFM-coupled four ½-spins, which has $H_{C1} = 3.7$ T and $H_{C2} = 7.4$ T with no slope in $0 < H < H_{C1}$, is presented by the black dashed line in the figure.

Considering the pseudospin-½ with the Yb $4f$ strong SOC, we should adopt a generalized magnetic exchange Hamiltonian[17–19,26–28] to explain the observed magnetization;

$$\mathcal{H}_{gen} = \sum_{i<j} \mathbf{S}_i \cdot \mathbf{J}_{ij} \cdot \mathbf{S}_j - \mu_B \mathbf{H} \cdot \sum_i \mathbf{g}_i \cdot \mathbf{S}_i. \quad (1)$$

The first and second terms represent the generalized magnetic exchange and Zeeman terms, respectively. The $i$th site pseudospin can be represented by the Pauli spin operator $\mathbf{S}_i$, and the exchange coupling $\mathbf{J}_{ij}$ between $\mathbf{S}_i$ and $\mathbf{S}_i$ is presented in a tensor form ($J^{\mu\nu}$). The Zeeman term is described with the Bohr magneton $\mu_B$, external magnetic field $\mathbf{H}$ and a g-tensor $\mathbf{g}_i$ reflecting the $i$th site single-ion magnetic anisotropy. The generalized Hamiltonian $\mathcal{H}_{gen}$ well explains observed INS results of the breathing pyrochlore $Ba_3Yb_2Zn_5O_{11}$ with only a few none-trivial $J^{\mu\nu}$ values as discussed below, and is effectively reduced to an effective exchange Hamiltonian with an additional DM interaction (antisymmetric exchange interaction) term to the conventional Heisenberg AFM Hamiltonian (Supplementary Note 1):

$$\mathcal{H}_{eff} = J \sum_{i<j} \mathbf{S}_i \cdot \mathbf{S}_j + \sum_{i<j} \mathbf{d}_{ij} \cdot (\mathbf{S}_i \times \mathbf{S}_j) - \mu_B \mathbf{H} \cdot \sum_i \mathbf{g}_i \cdot \mathbf{S}_i. \quad (2)$$

The first term represents the Heisenberg exchange interaction and the second term accounts for the DM interaction, which is a result of combined effects of the SOC and the superexchange interactions. According to the Moriya's rules[28,29], the DM vector $\mathbf{d}_{ij}$ does not disappear in $Ba_3Yb_2Zn_5O_{11}$ ($F\bar{4}3m$ space group) with no-inversion symmetry, and its direction (green arrows in the inset of Fig. 1c) is constrained by tetrahedron symmetry ($T_d$).

The step-like features in $M(H)$ at $H_{C1} = 3.5$ T and $H_{C2} = 8.8$ T reflect the spin singlet–triplet ($S_{eff} = 0$–1) and triplet–quintet ($S_{eff} = 1$–2) level crossings, respectively. This $M(H)$ curve can be well reproduced with the respective g-factor values $g_{||} = 3.0(1)$ and $g_\perp = 2.4(1)$ for parallel and perpendicular to the symmetry axis of each $YbO_6$, respectively (green line in Fig. 1d). These values are slightly larger than $g_{||} = 2.54$ and $g_\perp = 2.13$, which are more accurately determined from an electron paramagnetic resonance measurement (Supplementary Note 2). It is noticed that a hysteretic behaviour appears around $H_{C1}$. This behaviour reflects the Landau–Zener transition involving an avoided level crossing with an energy gap in a dissipative two-state model (inset of Fig. 1d)[30–33]. The level crossing energy gap hinders the adiabatic crossover between the singlet and triplet as the magnetic field increases or decreases across $H_{C1}$. Evidently, we observed that the hysteresis varies with the field sweep rate (Supplementary

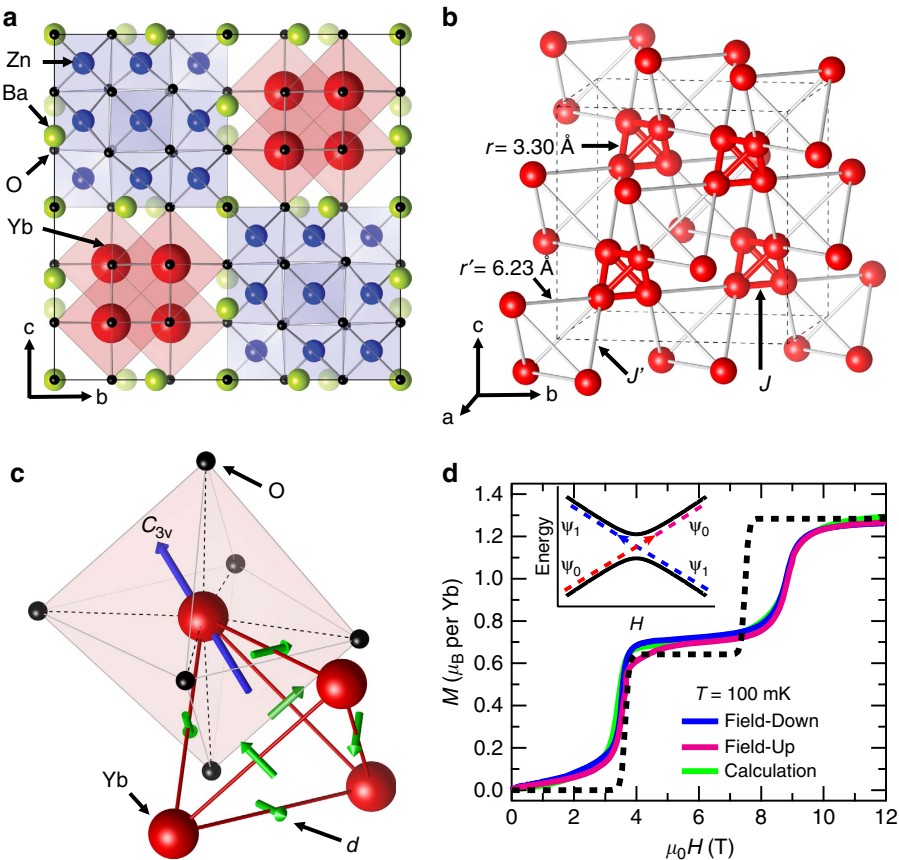

**Figure 1 | Crystallographic structure and magnetization curves of Ba$_3$Yb$_2$Zn$_5$O$_{11}$.** (**a**) Crystal structure of pyrocholre Ba$_3$Yb$_2$Zn$_5$O$_{11}$ with cubic space group $F\bar{4}3m$. Green, red, blue and black spheres represent Ba, Yb, Zn and O ions, respectivley. Two main blocks of Yb$_4$O$_{16}$ and Zn$_{10}$O$_{20}$ are alternated and Ba ions locate in the interstices. (**b**) Arrays of Yb ions and Yb$_4$ tetrahedrons with alternated Yb–Yb distances in the breathing pyrochlore structure. The inter-tetrahedron (red line) and intra-tetrahedron (grey line) Yb–Yb distances are $r = 3.30$ Å and $r' = 6.23$ Å, respectively. Corresponding magnetic exchange couplings are denoted by $J$ and $J'$. (**c**) Yb$_4$ tetrahedron and YbO$_6$ octahedron with trigonal distortion ($C_{3v}$). A blue arrow denotes $C_{3v}$ symmetry axis pointing along the [1 1 1] direction and green arrows indicate the DM vectors **d**'s determined from the Moriya's rule. (**d**) Field-dependent magnetization $M(H)$ measured for upfield (magenta) and downfield (blue) sweeps with a rate of 15 mT min$^{-1}$ at $T = 100$ mK, showing the level crossing critical fields of $H_{C1} = 3.5$ T and $H_{C2} = 8.8$ T. A green solid line displays adiabatic simulation results from $\mathcal{H}_{\text{eff}}$ with $J = 0.589$ meV, $d/J = 0.27$, $g_{\parallel} = 3.0$ and $g_{\perp} = 2.4$. A black dashed line displays simulation results at $T = 100$ mK with $H_{C1} = 3.7$ T and $H_{C2} = 7.4$ T from the conventional Heisenberg magnetic exchange Hamiltonian including the Zeeman term with an exchange coupling constant $J = 0.554$ meV and an isotropic $g$-factor $g = 2.569$ reported previously[24]. The inset is a schematic illustration of Landau – Zener transition between two energy states of $\psi_0$ and $\psi_1$ as in Fig. 2d. Adiabatic and non-adiabatic processes as a function of the external magnetic field are presented with solid and dashed lines, respectivley.

Note 3). At $H_{C2}$, the energy gap is enhanced, and the hysteretic feature becomes less effective. A finite slope in $M(H)$ can be also noticed in the range $0 < H < H_{C1}$. Due to the fact that the DM interaction admixes the singlet and triplet states, the ground state is no longer a pure $S_{\text{eff}} = 0$ state and the admixed triplet $S_{\text{eff}} = 1$ state contributes the weak field dependence to the magnetization[34]. While the anisotropic Zeeman term also can contribute to paramagnetic response to the applied field, its effect on $M(H)$ is less significant than that of DM in the breathing pyrochlore Ba$_3$Yb$_2$Zn$_5$O$_{11}$ (Supplementary Note 3).

**INS without a magnetic field.** To explore magnetic excitations in this novel quantum magnet, we performed INS measurements[20–22,35,36]. Figure 2a shows the intensities $I(Q,\omega)$ as a function of momentum and energy transfer obtained from the measurements at $T = 200$ mK and at the zero magnetic field. The $I(Q, \omega)$ exhibits four non-dispersive excitations, which correspond to the transitions from the $S_{\text{eff}} = 0$ ground state to the $S_{\text{eff}} = 1$ excited states. As temperature increases, the

$S_{\text{eff}} = 1$ states become partially occupied due to the thermal energy, and additional transitions from the $S_{\text{eff}} = 1$ states become available in the INS result. Indeed, we could observe additional non-dispersive excitations at 10 K as shown in Fig. 2b.

To confirm the molecular characteristics of the quantum spin state, we examined the $Q$-dependences of two dominant excitation intensities $I(Q)$, which are integrated over 0.45–0.6 meV and 0.65–0.8 meV regions in Fig. 2a. The obtained $I(Q)$ is compared with the calculated ones for an Yb$^{3+}$ single ion (black dashed line) and an Yb$_4$ tetrahedron molecule (green dashed line) as shown in Fig. 2c. The obtained $I(Q)$ is obviously well explained by the molecular model rather than the ionic model[35], supporting the presence of molecular quantum magnetism in Ba$_3$Yb$_2$Zn$_5$O$_{11}$. The $Q$ integrated intensities $I(\omega)$ presented in Fig. 2d are also well understood in a framework of the $S_{\text{eff}}$ molecular magnet states. At 200 mK, we identified four excitation peaks, while seven peaks are observable at 10 K, which could be indexed with 11 excitations. The corresponding excitations (vertical arrows) are described in the energy level

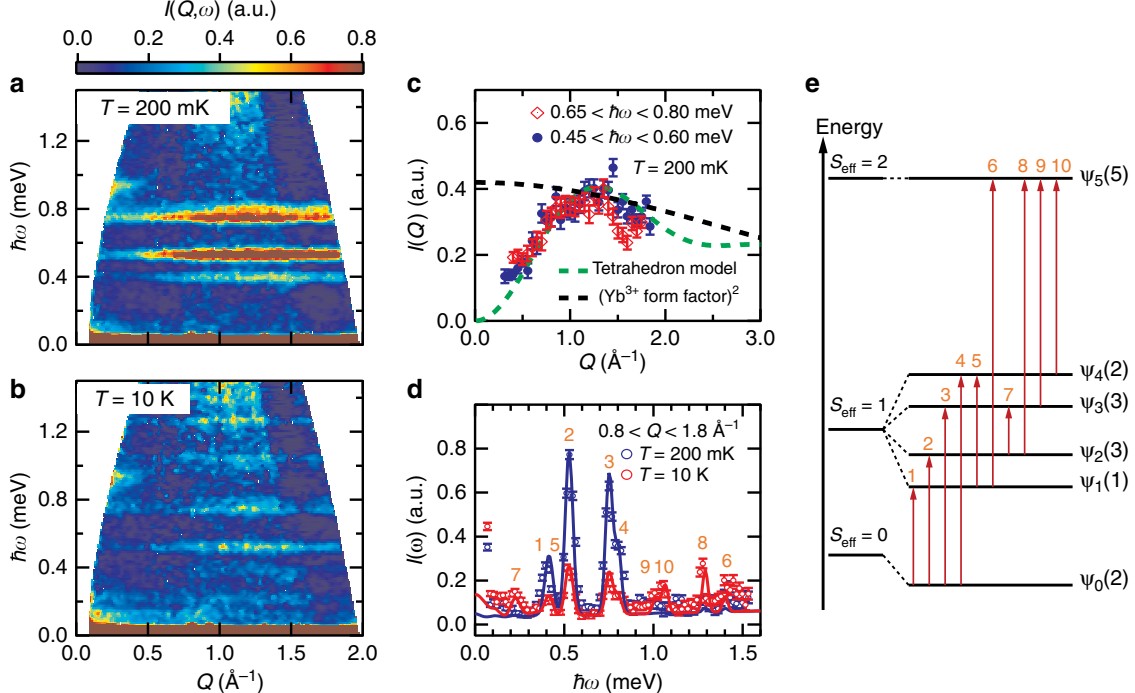

**Figure 2 | INS results and an energy diagram of a Yb$_4$ molecular tetrahedron.** (**a,b**) INS intensities $I(Q, \omega)$ as a function of momentum $Q$ and energy transfer $\hbar\omega$ measured using incident neutron energy $E = 2.27$ meV ( $= 6.0$ Å) at (**a**) $T = 200$ mK and (**b**) $T = 10$ K. (**c**) Constant-$\omega$ cuts $I(Q)$'s obtained by integrating over ranges of $0.45$ meV $< \hbar\omega < 0.60$ meV (blue filled circle) and $0.65$ meV $< \hbar\omega < 0.80$ meV (diamond) at 200 mK. A black dashed line represents the square of the Yb$^{3+}$ magnetic form factor, and a green dashed line does the model calcution for a Yb$_4$ tetrahedron, which expresses a functional form of $1 - \sin(Qr)/Qr$ at $r = 3.30$ Å. (**d**) Constant-$Q$ cuts $I(\omega)$'s obtained by integrating over a range of $0.8$ Å$^{-1} < Q < 1.8$ Å$^{-1}$ at $T = 200$ mK (blue circle) and $T = 10$ K (red circle). Solid lines present simulated $I(\omega)$'s from the effective Hamiltonian $\mathcal{H}_{\text{eff}}$. (**e**) Schematic energy level diagram extracted from diagonalization of $\mathcal{H}_{\text{eff}}$. The degeneracy of each energy level is presented in the paranthesis together with the eigenstate $\psi$. The excitations with indices from 1 to 10 (vertical red arrows) are observable in $I(\omega)$'s in **d**. Intensity error bars are square roots of intensities.

diagram based on the effective Hamiltonian $\mathcal{H}_{\text{eff}}$, which is schematically depicted in Fig. 2e.

The Heisenberg exchange splits total $2^4$ magnetic states of the Yb$_4$ molecule, consisting of four ½ pseudospins, into doubly degenerated $S_{\text{eff}} = 0$, triply degenerated $S_{\text{eff}} = 1$ and non-degenerated $S_{\text{eff}} = 2$ states. Due to the DM interaction, the $3 \times 3$ $S_{\text{eff}} = 1$ states are admixed with the lowest $S_{\text{eff}} = 0$ states and split into four states, $\psi_1(1)$, $\psi_2(3)$, $\psi_3(3)$ and $\psi_4(2)$, with degeneracies presented in the parenthesis, while the ground state $\psi_0(2)$ is further lowered in energy. On the other hand, the $S_{\text{eff}} = 2$ quintet state $\psi_5(5)$ is not affected. For detailed analyses, we calculated eigenstates and eigenvalues of $\mathcal{H}_{\text{eff}}$ using the exact diagonalization method, and estimated the magnetic scattering cross-sections. With optimized values of $J = 0.589$ meV and $d = 0.158$ meV (refs 37,38), we obtained simulated $I(\omega)$ spectra at 200 mK (blue line) and 10 K (red line), which well reproduce the experimental data as shown in Fig. 2d. The obtained large DM value ($d/J = 0.27$), which reflects the strong SOC of Yb$^{3+}$ ions, is consistent with the value theoretically estimated for the Yb–O–Yb superexchange hopping (Supplementary Note 4).

**INS with magnetic fields.** Validity of the effective Hamiltonian can be also confirmed in the field-dependent INS results at 200 mK. Figure 3a shows $I(\omega)$, integration of $I(Q,\omega)$ over $0.8$ Å$^{-1} < Q < 1.8$ Å$^{-1}$, under various external magnetic fields in comparison with the theoretical simulations. The excitation peaks evolve with the external field. The simulated $I(\omega)$ spectra represent spherically averaged $I(\omega)$ from exactly diagonalized $\mathcal{H}_{\text{eff}}$. The simulations well reproduce the experiments with the g-factor

values $g_{\parallel} = 2.62(2)$ and $g_{\perp} = 2.33(2)$, which are slightly smaller than the values obtained from the $M(H)$ curve, likely due to an estimation error. These g-factor values are also consistent with the values estimated from the electron paramagnetic resonance spectrum and those determined from YbO$_6$ CF analyses for reported high-energy neutron excitation spectra[25] as discussed in Supplementary Notes 2 and 4, respectively.

Figure 3b shows the calculated excitation energy diagrams as a function of external magnetic field $H$ along the principle axes [0 0 1], [1 1 0] and [1 1 1] using the obtained values of the $J$, $d$ and g-factors. The colour of the excitation energy line represents the magnetization value $<gS^z>$ ranged from $-g_{\parallel}/2$ to $+g_{\parallel}/2$ as presented by a colour scale bar in Fig. 3c. One can notice that the overall energy diagram is almost identical for the three axes except minor variations in the excitation energies, which appear as peak broadenings in the observed $I(\omega)$ in Fig. 3a, and the calculated energies coincide with the peak positions. We also trace three excitation peaks as marked in Fig. 3a,b. The first excitation (diamond) corresponds to $\psi_0$ ($S_{\text{eff}} \approx 0$) $\rightarrow \psi_1$ ($S_{\text{eff}} \approx 1$) for $H < H_{\text{C1}}$ ($\approx 3.5$ T). For $H > H_{\text{C1}}$, the Zeeman energy of $\psi_1$ overcomes their zero-field energy difference, that is, level crossing, and the excitation is switched to the second one (star), $\psi_1 \rightarrow \psi_0$. As $H$ exceeds $H_{\text{C2}}$, $\psi_5$ ($S_{\text{eff}} = 2$) becomes the ground state, and the third one (triangle) representing the excitation $\psi_1 \rightarrow \psi_5$ becomes unavailable. The avoided level crossing feature around $H = H_{\text{C1}}$ is also examined in the calculated excitation. Figure 3c shows the $\psi_0 \rightarrow \psi_1$ to $\psi_1 \rightarrow \psi_0$ excitation crossover for $H//[1\ 1\ 0]$ in a very-low-energy region. Even though $H$ approaches $H_{\text{C1}}$, the excitation energy between $\psi_0$ and $\psi_1$ does not vanish, confirming existence of a finite gap, that is, avoided

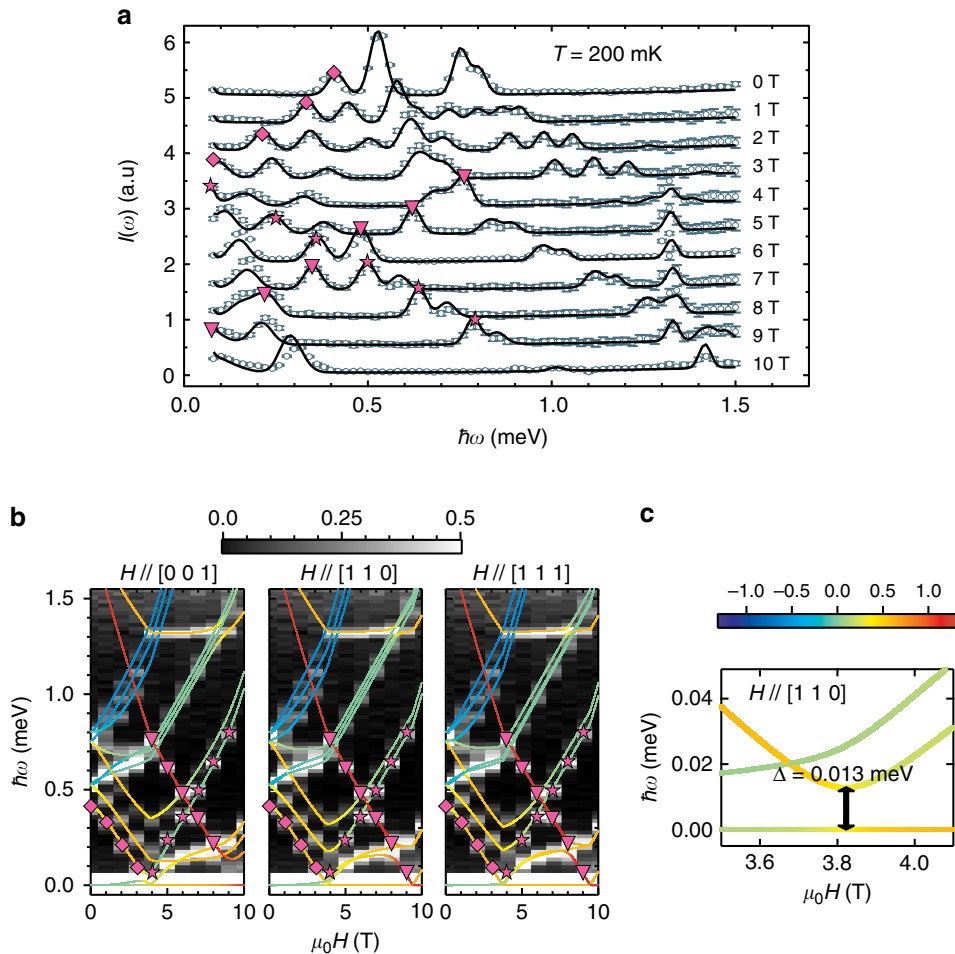

**Figure 3 | Field-dependent INS spectra and evolution of excitation energies.** (**a**) Constant-$Q$ cuts $I(\omega)$'s obtained by integrating measured $I(Q,\omega)$'s over a range of $0.8\,\text{Å}^{-1} < Q < 1.8\,\text{Å}^{-1}$ at $T = 200\,\text{mK}$ in external magnetic fields ranging from $H = 0$ to $H = 10\,\text{T}$. Solid lines are obtained by fitting the experimental $I(\omega)$'s with parameters of $J$, $d$ and $g$-factors in $\mathcal{H}_{\text{eff}}$. (**b**) Calculated excitation energy diagrams as a function of the $H$-field along the three principal axes [0 0 1], [1 1 0] and [1 1 1] are overlapped on the contour map of $I(\omega)$'s. Excitation peak positions of $I(\omega)$'s agree well with calculations. Peak intensites are presented by the grey scale bar. Selected three excitations, related to the level crossings at $H_{\text{C1}}$ and $H_{\text{C2}}$ are, respectively, marked with stars, diamonds and triangles in **a** and **b**. (**c**) Avoided level crossing along $H//$[1 1 0] near $H = H_{\text{C1}}$ in a low-energy region. The colour scale bar represents the magnetization value $<\mathbf{g}S^z>$ per Yb for the line colours in **b** and **c**. On the other hand, the avoided level crossing feature appears along $H//$[0 0 1] near $H = H_{\text{C2}}$. Intensity error bars are square roots of intensities.

level crossing, which originates the hysteretic behaviour of $M(H)$ discussed above. The estimated gap energy $\Delta \approx 0.013\,\text{meV}$ corresponds to a gigahertz range in a qubit model system.

## Discussion

Our analysis demonstrates that the inorganic solid $Ba_3Yb_2Zn_5O_{11}$ realizes a novel molecular quantum magnet. By virtue of strong SOC of Yb $4f$ electrons, each $Yb^{3+}$ ion has a spin–orbit coupled pseudospin-½, and the antisymmetric DM exchange interaction plays a crucial role in the magnetism. This quantum magnetism exhibits not only the exotic Landau–Zener transition involving avoided level crossing but also paramagnetic responses under a magnetic field that differ significantly from conventional molecular magnetism. In this inorganic material, the inter-molecular exchange coupling $J'$ is negligible, but as the inter-molecular spacing is reduced by the substitution of non-magnetic ions with smaller ionic sizes or by applying pressure, $J'$ increases to turn on the entanglement between the molecular spins. A weakly coupled alternating pyrochlore system can be considered as a protocol for quantum gates and spin manipulations by electric fields, as proposed for weakly coupled molecular spin triangles[39]. When $J'$ becomes significant, the system ends to be a typical frustrated magnet[23]. Therefore, this material provides a promising starting point for exploration of an uncharted crossover from molecular to entangled quantum magnetism.

## Methods

**Sample synthesis and magnetization.** Polycrystalline $Ba_3Yb_2Zn_5O_{11}$ samples were prepared by the solid-state reaction method from a stoichiometric mixture of $Ba_2CO_3$ (99.999%), $Yb_2O_3$ (99.99%) and ZnO (99.999%) powders as in the (ref. 24). The mixture pellet was successively sintered at 1,000 °C for 24 h and at 1,120 °C for 24 h in air. The isothermal magnetization was determined for a 60 mg of pelletized sample below $T = 1.8\,\text{K}$ using a conventional Faraday force magnetometer, which measures changes in the electric capacitance induced by a magnetic field gradient. To avoid any movement of grains during the measurements, we pelletized a 60 mg of powder to a cubic-shaped hard solid, and firmly mounted it on the magnetometer sample holder, and no crack was observable on the pellet after the measurements. The measurements were performed under a static magnetic field in a range from 0 to 12 T with various sweep rates from 7.5 to 30 mT min⁻¹. The magnetization at 1.8 K, which was measured using the Quantum Design's Magnetic Property Measurement System, was utilized as a reference to determine the magnetizations at different temperatures.

**Elastic and inelastic neutron scattering.** Neutron powder diffraction experiment was conducted at the high-resolution powder diffraction beamline in HANARO for

the structural information (Supplementary Fig. 1; Supplementary Table 1). Powder samples of 5 g were sealed in vanadium container and placed on a closed cycle refrigerator for the diffraction measurements at 4 K. The crystal structure was determined from the Rietveld refinement using the FullProf suite software[40]. Time-of-flight neutron scattering experiment was carried out at the Disk Chopper Spectrometer beamline in the National Institute of Standards and Technology Center for Neutron Research. The neutron scattering data were obtained at $T = 200$ mK and 10 K. The incident neutron energy was set to be 2.27 meV ($= 6$ Å) with an energy resolution of 64 μeV at the elastic line. A polycrystalline sample of 10 g was sealed in a copper container, and inserted into a 10 T vertical field magnet equipped with a dilution refrigerator.

The inelastic magnetic neutron scattering intensity for isolated $Yb_4$ tetrahedrons is given by

$$I(\mathbf{Q}, \omega) = I_0 \frac{k_f}{k_i} F^2(Q) \sum_{\alpha, \beta} \left( \delta_{\alpha\beta} - \frac{Q_\alpha Q_\beta}{Q^2} \right) S^{\alpha\beta}(\mathbf{Q}, \omega), \quad (3)$$

where $I_0$ is a scale factor, $k_i$ and $k_f$ are initial and final neutron wave vectors, and $\mathbf{Q}$ and $\omega$ are the momentum and energy transfers, respectively. $F(Q)$ is a dimensionless magnetic form factor of $Yb^{3+}$, and $\alpha, \beta = x, y$ and $z$, and the dynamical structure factor $S^{\alpha\beta}(\mathbf{Q}, \omega)$ is defined as

$$S^{\alpha\beta}(\mathbf{Q}, \omega) = \sum_{i,i'} e^{i\mathbf{Q}\cdot(\mathbf{r}_i - \mathbf{r}_{i'})} \sum_{\psi, \psi'} P_\psi \langle \psi | M_{i,\alpha} | \psi' \rangle \langle \psi' | M_{i',\beta} | \psi \rangle \delta(\hbar\omega + E_\psi - E_{\psi'}),$$
$$(4)$$

where $i$ is the site index and $\psi$ is an eigenstate $|\psi\rangle$ of $\mathcal{H}_{\text{eff}}$ with an energy $E_\psi$. $P_\psi$ denotes the thermal population $\exp(-E_\psi/k_B T)/Z$ with the partition function $Z$, and $\mathbf{M}_i$ is the $i$-site effective magnetic moment $g_\perp (S_{i,x_i} \hat{\mathbf{x}}_i - S_{i,y_i} \hat{\mathbf{y}}_i) + g_\parallel S_{i,z_i} \hat{\mathbf{z}}_i$. For a powder sample with an external magnetic field $\mathbf{H}$, the inelastic magnetic neutron scattering intensity is obtained by averaging $I(\mathbf{Q}, \omega)$ over all directions of $\mathbf{Q}$[41] and $\mathbf{H}$ as

$$I(Q, \omega) = \int \frac{d\hat{\mathbf{H}}}{4\pi} \int \frac{d\hat{\mathbf{Q}}}{4\pi} I(\mathbf{Q}, \omega), \quad (5)$$

which is adopted for the INS data fitting.

**Code availability.** We declare that the data-simulation code supporting the findings of this study are available within article's Supplementary Information file (Supplementary Software 1).

**Data availability.** The data that support the findings of this study are available from the corresponding author on request.

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

## Acknowledgements

We became aware of the experimental works by Rau et al.[37] and Haku et al.[38] after we have submitted this paper. They also obtained the consistent results with ours using INS without an external magnetic field and the same model Hamiltonian. In our work, we have advanced by showing that the DM interaction associated with the avoided energy level crossing plays a crucial role for unconventional quantum magnetic behaviours such as a hysteresis and paramagnetic response in magnetization, which is confirmed by INS with an external magnetic field. We are grateful to K.-B. Lee and K.-S. Park for enlightning discussions. This work is supported by the National Research Foundation (NRF) through the Ministry of Science, ICP & Future Planning (MSIP) (No. 2016K1A4A4A01922028). B.H.K. is supported by the RIKEN iTHES Project. S.-H.D. and K.-Y.C. are supported by the Korea Research Foundation grant (No. 2009-0076079) funded by the Korea government (MEST). S.L. is supported by the NRF under the contract NRF-2012M2A2A6002461.

## Author contributions

S.-Y.P., N.P.B. and S.J. performed INS measurement. J.-H.K., D.J. and M.B. carried out magnetization measurement. S.-Y.P., S.L. and S.J. performed neutron powder diffraction

measurement. S.-H.D. and K.-Y.C. synthesized samples. S.-Y.P., B.-H.K., B.S. and S.J. developed the theoretical model of the effective Hamiltonian. D.-H.K., B.-H.K. and J.-H.P. performed CF and superexchange model calculations. S.-Y.P. and S.J. analysed the data. S.-Y.P., S.J. and J.-H.P. wrote the manuscript. J.-H.P. and S.J. initiated and supervised the research.

## Additional information

**Competing financial interests:** The authors declare no competing financial interests.

