## [Peer Review File · Nature Communications]

Reviewers' comments:

Reviewer #1 (Remarks to the Author):

The manuscript provides a detailed account of the magnetic excitations in the material $\text{Ba}_3\text{Yb}_2\text{Zn}_5\text{O}_{11}$, in comparison to a molecular magnetic model composed of four (pseudo)-spin 1/2 objects. The Yb^{3+} magnetic sub-lattice is what is commonly referred to as "breathing pyrochlore", which naturally provides tetrahedral subunits from which to form such a 4-spin molecular state. The calculations are done by considering the CEF effects on the Yb^{3+} ion to account for the pseudo-spin 1/2 states, and then constructing a symmetry-allowed anisotropic exchange Hamiltonian with fitted exchange constants, which is then diagonalized for 4 sites. The results are in nearly perfect agreement with the inelastic neutron scattering data, in both zero field and applied magnetic field, which is quite impressive. Thus, it does appear that the system is well understood by the single tetrahedron model. The lattice structure is such that pressure could be used to increase interactions between tetrahedra which could show the crossover between molecular magnetism and frustrated magnetism. I believe in this context, this work gives a very valuable scientific result on the "starting point" of that potential crossover.

Good agreement is also found between the proposed model and the M vs. H curves, which shows the two expected plateaus associated with level crossings from $S_{\text{tot}} = 0$ to $S_{\text{tot}} = 1$ and then to $S_{\text{tot}} = 2$. However, there is some hysteresis observed in the M vs. H curves and the authors discuss a Landau-Zener picture of non-adiabatic transitions across an avoided crossing in order to understand this. Actually, I wonder whether this hysteresis could be caused by grain reorientation in the applied field. This is a known effect when magnetic fields are applied to polycrystalline samples. The authors could check this by repeating the M vs. H measurement with a small amount of wax encasing the powder sample (e.g. eicosane wax).

While overall this work is very well done, and the manuscript is overall well written (there are some clarifications needed as detailed below), unfortunately the authors have failed to make reference to an extremely similar result on the same compound, which was posted as a preprint earlier this year (see Rau et al, arXiv:1601.04104 [cond-mat.str-el] (2016)). Many of the same techniques are applied and the same results are obtained. I feel that this preprint must be acknowledged and referenced in the present manuscript. Perhaps the authors could comment on what aspects of their work go above and beyond the other preprint (for starters, in the present work the effect of an applied field on the spin excitations is investigated); this may help the editors decide whether the novelty of this manuscript is really high enough for Nature Communications.

Below I list some needed clarifications to the manuscript:

Main text:

- The behavior of each tetrahedron is likened to a " $S_{\text{eff}} = 2$ state" (on line 29, line 108). However, the $S_{\text{eff}}=0$ and $S_{\text{eff}}=1$ states appear first at lower field. See also line 64 which explains it this way. This needs clarification.
- What the authors mean by "shifts of critical fields" is not clear to me (line 65)
- the $F-43m$ space group does have an inversion center; authors should clarify on line 81 that it is the nearest neighbor Yb-Yb "bonds" which do not have an inversion center, leading to the possibility of a DM interaction.
- It should be stated clearly in the main text that the sample is polycrystalline, since an applied field is used. Currently it is only stated in the methods section at the end and this could be misleading.

- On line 67, the authors should emphasize that only for the specific values of the J-tensor elements obtained from fitting, the Hamiltonian can be reduced to the Heisenberg plus DM form shown. The way it is currently written makes it sound like a more general property of the model.

Supplementary information:

- there is a typo in the definition of "g_perp" in the SI, section 1
- Could the authors confirm that in Heff in the supplemental information (section 1), the S_i and S_j operators are decomposed into the CEF ground doublet? Or are they Pauli spin matrices? This is important for reproducing neutron scattering intensities.

In summary, while I find the work interesting and well done, at the present time I cannot recommend publication. There are a number of clarifications needed (see above) and, most importantly, a prior work that produces many of the same results is not discussed. I would recommend publication once these issues are addressed.

Reviewer #2 (Remarks to the Author):

A. This manuscript presents the magnetism of the rather novel compound $Ba_3Yb_2Zn_5O_{11}$. The main novelty is the existence of separated molecule like tetrahedral units. Another interesting feature is the astonishingly large DM interaction of roughly 1/4 of the Heisenberg interaction, which is due to the strong spin orbit interaction.

B. The investigations are novel, the authors suggest usage in quantum computing due to the existence of voided level crossings.

C. The experimental data are convincing and of high quality. The interpretation of some parts suffers from some systematic problems:

- The authors speak of a $S=2$ system, which is totally wrong. The four $s=1/2$ spins are coupled antiferromagnetically and constitute an $S=0$ ground state. A $S=2$ system would possess a different magnetization curve and a different level scheme.
- The magnetization curve shows as theory the result for a Heisenberg model, which as the authors state in all other discussions, is inappropriate. It is not clear to me why the authors do not show a full diagonalization study of the magnetization since they calculate everything for the INS spectra with the appropriate Hamiltonian.
- I would say that magnetically the system is not a pyrochlore since it is not corner sharing, as misleadingly stated in line 50. It is a system of tetrahedra which interact through bonds between corners.

D. The authors claim that the inter-tetrahedral interaction is negligible, since the distance is 6.23 Angstrom. Is there any further evidence? Could a weak inter-tetrahedral interaction lead to a similar magnetic behavior?

E. I think the manuscript needs revision, see above, especially regarding clarity of phrases and theory of magnetization.

Reviewer #3 (Remarks to the Author):

A. The paper reports some very attractive data measured on a solid state material where Yb_4 tetrahedra are isolated within the lattice. This is a very good idea, and there is some beautiful

measurements, including field dependent INS data.

B. The results are interesting, but the authors have missed some significant work in the past that should be discussed and they should read and consider before submitting a revision.

Very specifically: there is beautiful work from the Güdel group on use of INS to study lanthanide dimers, including Yb₂ dimers. Relevant papers include: *Inorganic Chemistry*, 1990, 29, 4081; *Inorg. Chem.*, 1994, 33, 1133; *PRL*, 1989, 62, 210.

There is also a recent paper in *Nature Communications*, 2014, 5, 5243 which discusses exchange interactions in a dysprosium dimer, and which modelled data in a very different way to that proposed here.

C. The data are very good I think. I am far less convinced about the Hamiltonian used and the way data are modelled. Specifically: treating this exchange with an isotropic exchange interaction and then introducing a DM interaction to obtain anisotropy seems odd. I think the authors should have used an anisotropic exchange interaction from the start (as in the *Nature Comm* paper referenced above). At the very least the authors should try this approach and show that it does not work or is no better.

D. Statistics seem fine.

E. I would like an explanation of why they used the Hamiltonian they used before considering whether conclusions make sense.

F. There are two aspects missing. I think there should be an EPR study especially as the g-values used for the techniques modelled vary. I don't understand at all the explanation of the variation in g-values. My guess is the variation is due to the wrong Hamiltonian being used.

Secondly, a CASSCF calculation of the g-values of the Yb sites should be reasonably robust in this case. I think some type of electronic structure calculation would add to the paper significantly.

G. The referencing is weak. Many of the papers referenced are irrelevant, but the very large body of work that has appeared on lanthanide single molecules is largely ignored. The Chibotaru work on Dy₃ triangles is certainly relevant here. Also, previous field dependent INS studies are not referenced (e.g. *PRL*, 2007, 98, 167401). This is easily rectified, and an easy mistake to make when moving into a new area.

H. The paper is well written and clear. I don't like Figure 3, which is poor and improvements are needed. I just don't see the relationship between Figure 3a and 3b. In Figure 3a, the energy window observed by INS appears to be moving with field; I don't understand why that would be the case. While the 3D-plot looks attractive, it might be easier for a reader if these data were more simply represented against a common energy axis and with the data for each field plotted against the same window.

Figure 3b is rather difficult to understand and I don't think the black background helps. It is not even obvious which transitions are observed. Again, given the model calculated is anisotropic, why is it based on an isotropic exchange interaction plus DM, rather than an anisotropic exchange? At the very least the authors need to demonstrate that an anisotropic exchange interaction would give a less good fit. As the INS was measured on a powder, why give the INS calculated for three principle directions?

I think the data already reported could form the basis of a paper in *Nature Communications*, if the authors did some more experiments (EPR spectroscopy, CASSCF calculations), read and referenced some relevant previous work, and improved Figure 3.

I'm afraid my review reads more negatively than I would like: I stress - the data are very good and really interesting.

Reviewer #1 (Remarks to the Author):

The manuscript provides a detailed account of the magnetic excitations in the material $\text{Ba}_3\text{Yb}_2\text{Zn}_5\text{O}_{11}$, in comparison to a molecular magnetic model composed of four (pseudo)-spin 1/2 objects. The Yb^{3+} magnetic sub-lattice is what is commonly referred to as "breathing pyrochlore", which naturally provides tetrahedral subunits from which to form such a 4-spin molecular state. The calculations are done by considering the CEF effects on the Yb^{3+} ion to account for the pseudo-spin 1/2 states, and then constructing a symmetry-allowed anisotropic exchange Hamiltonian with fitted exchange constants, which is then diagonalized for 4 sites. The results are in nearly perfect agreement with the inelastic neutron scattering data, in both zero field and applied magnetic field, which is quite impressive. Thus, it does appear that the system is well understood by the single tetrahedron model. The lattice structure is such that pressure could be used to increase interactions between tetrahedral which could show the crossover between molecular magnetism and frustrated magnetism. I believe in this context, this work gives a very valuable scientific result on the "starting point" of that potential crossover.

Good agreement is also found between the proposed model and the M vs. H curves, which shows the two expected plateaus associated with level crossings from $S_{\text{tot}} = 0$ to $S_{\text{tot}} = 1$ and then to $S_{\text{tot}} = 2$. However, there is some hysteresis observed in the M vs. H curves and the authors discuss a Landau-Zener picture of non-adiabatic transitions across an avoided crossing in order to understand this. Actually, I wonder whether this hysteresis could be caused by grain reorientation in the applied field. This is a known effect when magnetic fields are applied to polycrystalline samples. The authors could check this by repeating the M vs. H measurement with a small amount of wax encasing the powder sample (e.g. eicosane wax).

Response 1-1: To avoid any independent grain movements, we performed the magnetization measurements on a pelletized solid sample, not the original powder form samples, as described in the method section. After the measurements, however, we did not recognize any change or damage on it, and could confidently discard any possibility of independent grain movements during the measurements.

We modified the method section to clear out this issue in the revised manuscript.

While overall this work is very well done, and the manuscript is overall well written (there are some clarifications needed as detailed below), unfortunately the authors have failed to make reference to an extremely similar result on the same compound, which was posted as a preprint earlier this year (see Rau et al, arXiv:1601.04104 [cond-mat.str-el] (2016)). Many of the same techniques are applied and the same results are obtained. I feel that this preprint must be acknowledged and referenced in the present manuscript. Perhaps the authors could comment on what aspects of their work go above and beyond the other preprint (for starters, in the present work the effect of an applied field on the spin excitations is investigated); this may help the editors decide whether the novelty of this manuscript is really high enough for Nature Communications.

Response 1-2: We thank the reviewer. Unfortunately, we were aware of the experimental work (zero-field INS results only) by Rau *et al.*, arXiv:1601.04104 [cond-mat.str-el] (2016) after our first submission. Now we acknowledged and referenced this work in the revised manuscript as recommended by the reviewer.

Below I list some needed clarifications to the manuscript:

Main text:

- The behavior of each tetrahedron is likened to a " $S_{\text{eff}} = 2$ state" (on line 29, line 108). However, the $S_{\text{eff}}=0$ and $S_{\text{eff}}=1$ states appear first at lower field. See also line 64 which explains it this way. This needs clarification.

Response 1-3: As pointed out by the reviewer, the true ground state of this AFM coupled pseudospin- $\frac{1}{2}$ Yb_4 molecule is the $S_{\text{eff}} = 0$ state. Meanwhile the $S_{\text{eff}} = 1$ and 2 becomes the lowest states as the external magnetic field increases across H_{C1} and H_{C2} , respectively. This point is now clearly described in the revised manuscript.

- What the authors mean by "shifts of critical fields" is not clear to me (line 65)

Response 1-4: "shifts of critical fields" means that observed two critical fields, $H_{C1} = 3.5$ T and $H_{C2} = 8.8$ T, differ from the simulated $H_{C1} = 3.7$ T and $H_{C2} = 7.4$ T by the simple Heisenberg Hamiltonian.

To express the precise meaning of "shifts of critical fields", values of critical fields are added in the revised manuscript.

- the F-43m space group does have an inversion center; authors should clarify on line 81 that it is the nearest neighbor Yb-Yb "bonds" which do not have an inversion center, leading to the possibility of a DM interaction.

Response 1-5: The reviewer seems to be somewhat confused by the '-4' symmetry element in the F-43m, which represents an improper rotation (4-fold rotoinversion) symmetry, rather than the ordinary inversion symmetry. More specifically, the point group (T_d) of F-43m space group consists only of E , $3C_2$, $8C_3$, $6S_4$, $6\sigma_d$ symmetry elements, but no inversion symmetry element I (*Internal Tables for Crystallography. Vol. A: Space-group symmetry*, Springer Netherlands (2002)).

- It should be stated clearly in the main text that the sample is polycrystalline, since an applied field is used. Currently it is only stated in the methods section at the end and this could be misleading.

Response 1-6: As suggested by the reviewer, the polycrystalline sample is now stated in the main text.

- On line 67, the authors should emphasize that only for the specific values of the J-tensor elements obtained from fitting, the Hamiltonian can be reduced to the Heisenberg plus DM form shown. The way it is currently written makes it sound like a more general property of the model.

Response 1-7: As suggested by the reviewer, the corresponding statements are modified to clarify that the INS data require only a few none-trivial **J**-tensor elements, and that the general Hamiltonian is effectively reduced to the simplified effective Hamiltonian in this specific system.

Supplementary information:

- there is a typo in the definition of "g_perp" in the SI, section 1

Response 1-8: Thanks. It is now corrected.

- Could the authors confirm that in Heff in the supplemental information (section 1), the S_i and S_j operators are decomposed into the CEF ground doublet? Or are they Pauli spin matrices? This is important for reproducing neutron scattering intensities.

Response 1-9: As we described in Supplementary I, the pseudospin- $1/2$ operators \mathbf{S}_i , which represents the spin-orbit coupled total angular momentum rather than the conventional electron spin, has the same forms of the Pauli spin matrices in the subspace of the Kramers doublet, $|D_+\rangle_{z_i}$ and $|D_-\rangle_{z_i}$. The pseudo spin operators \mathbf{S}_i in the effective Hamiltonian and their matrix representations are given by

$$\begin{aligned} S_{i,x_i} &= \frac{1}{2} (|D_+\rangle_{z_i z_i} \langle D_-| + |D_-\rangle_{z_i z_i} \langle D_+|) \doteq \frac{1}{2} \begin{pmatrix} 0 & 1 \\ 1 & 0 \end{pmatrix}, \\ S_{i,y_i} &= \frac{1}{2i} (|D_+\rangle_{z_i z_i} \langle D_-| - |D_-\rangle_{z_i z_i} \langle D_+|) \doteq \frac{1}{2i} \begin{pmatrix} 0 & 1 \\ -1 & 0 \end{pmatrix}, \text{ and} \\ S_{i,z_i} &= \frac{1}{2} (|D_+\rangle_{z_i z_i} \langle D_+| - |D_-\rangle_{z_i z_i} \langle D_-|) \doteq \frac{1}{2} \begin{pmatrix} 1 & 0 \\ 0 & -1 \end{pmatrix}, \end{aligned}$$

with ${}_{z_i} \langle D_+ | D_+ \rangle_{z_i} = {}_{z_i} \langle D_- | D_- \rangle_{z_i} = 1$ and ${}_{z_i} \langle D_+ | D_- \rangle_{z_i} = {}_{z_i} \langle D_- | D_+ \rangle_{z_i} = 0$.

Thus the magnetic moment, which is conventionally defined to be $g\mathbf{J}$, is now presented by a vector form of operator $\mathbf{M}_i = g_{\perp} (S_{i,x_i} \hat{\mathbf{x}}_i - S_{i,y_i} \hat{\mathbf{y}}_i) + g_{\parallel} S_{i,z_i} \hat{\mathbf{z}}_i$ with effective anisotropic g -factors g_{\perp} and g_{\parallel} , and the neutron intensity was estimated for \mathbf{M}_i as described in the Supplementary III.

In summary, while I find the work interesting and well done, at the present time I cannot recommend publication. There are a number of clarifications needed (see above) and, most importantly, a prior work that produces many of the same results is not discussed. I would recommend publication once these issues are addressed.

Response 1-10: We addressed all the issues raised by the reviewer in the rebuttal letter, and modified the manuscript accordingly. We believe that the manuscript becomes clearer and stronger. Thanks the reviewer for valuable criticisms and recommendations.

Reviewer #2 (Remarks to the Author):

A. This manuscript presents the magnetism of the rather novel compound $\text{Ba}_3\text{Yb}_2\text{Zn}_5\text{O}_{11}$. The main novelty is the existence of separated molecule like tetrahedral units. Another interesting feature is the astonishingly large DM interaction of roughly $1/4$ of the Heisenberg interaction, which is due to the strong spin orbit interaction.

B. The investigations are novel, the authors suggest usage in quantum computing due to the existence of voided level crossings.

C. The experimental data are convincing and of high quality. The interpretation of some parts suffers from some systematic problems:

Response 2-1: The reviewer seems to agree that our works are novel and valuable and that the data are convincing and of high quality although she/he raised some questions. All the criticisms and questions are addressed below and the manuscript is modified accordingly.

- The authors speak of a $S=2$ system, which is totally wrong. The four $s=1/2$ spins are coupled antiferromagnetically and constitute an $S=0$ ground state. A $S=2$ system would possess a different magnetization curve and a different level scheme.

Response 2-2: We agree that our previous description of “the $S_{\text{eff}} = 2$ molecular magnet state” is rather confusing. The true ground state of this AFM coupled pseudospin- $1/2$ Yb_4 molecule is the $S_{\text{eff}} = 0$ state. Meanwhile the $S_{\text{eff}} = 1$ and 2 becomes the lowest states as the external magnetic field increases across H_{C1} and H_{C2} , respectively. This point is now clearly described in the revised manuscript.

- The magnetization curve shows as theory the result for a Heisenberg model, which as the authors state in all other discussions, is inappropriate. It is not clear to me why the authors do not show a full diagonalization study of the magnetization since they calculate everything for the INS spectra with the appropriate Hamiltonian.

Response 2-3: We also calculated the magnetization curve using the exact diagonalization with the appropriate Hamiltonian \mathcal{H}_{eff} as for the INS spectra analyses, but the resulting adiabatic magnetization curve was only presented in Supplementary II (Fig. S2). For the complete comparison, this magnetization curve is also implemented together with the simplified result for the Heisenberg model in Fig. 1d of the revised manuscript.

- I would say that magnetically the system is not a pyrochlore since it is not corner sharing, as misleadingly stated in line 50. It is a system of tetrahedra which interact through bonds between corners.

Response 2-4: The system $\text{Ba}_3\text{Yb}_2\text{Zn}_5\text{O}_{11}$ is indeed crystalized in a pyrochlore structure with the corner shared Yb_4 tetrahedra as also stated in the previous reports (Ref. 18: Kimura *et al.* Phys. Rev. B **90**, 060414 (2014) and Ref. 17: Savary *et al.* arXiv:1511.06972). The reviewer seems to be confused, likely due to a rather vague figure presentation of the crystal structure presented in Fig. 1b of the original manuscript. Thus, we modified the Fig. 1b in the manuscript, which now gives a clear display of the corner shared tetrahedra.

D. The authors claim that the inter-tetrahedral interaction is negligible, since the distance is 6.23 Angstrom. Is there any further evidence? Could a weak inter-tetrahedral interaction lead to a similar magnetic behavior?

Response 2-5: Let us consider a similar magnetic pyrochlore systems $\text{LiInCr}_4\text{O}_8$ with the Cr-Cr distance ratio of the inter- to intra-tetrahedron of $3.05 \text{ \AA}/2.90 \text{ \AA}$. The inter-tetrahedron exchange coupling J' is reported to be only 10% of the intra-tetrahedron one J , i.e. $J'/J = 0.1$. Meanwhile, the Yb-Yb distance ratio in $\text{Ba}_3\text{Yb}_2\text{Zn}_5\text{O}_{11}$ is as big as $6.23 \text{ \AA}/3.30 \text{ \AA}$, and thus the J'/J is expected to be even much less than a few %. In the experimental point of view, we could well reproduce both the macroscopic (magnetization) and microscopic (Q -dependent INS spectra) data in the framework of the single Yb_4 molecular Hamiltonian without the inter-tetrahedral interaction as shown in Fig. 1d and Fig. 2c, respectively. If the inter-tetrahedral interaction becomes considerable, the magnetic excitation will be dispersive and its effects will disturb considerably the microscopic and/or macroscopic data. However, the magnetic excitation is non-dispersive and the effects were not observable within the instrumental resolution limit down to 100 mK in both the macroscopic and microscopic results.

E. I think the manuscript needs revision, see above, especially regarding clarity of phrases and theory of magnetization.

Response 2-6: We agree that some phrases and descriptions are vague or incorrect. We thank the reviewer for valuable comments, and modified the manuscript accordingly.

Reviewer #3 (Remarks to the Author):

A. The paper reports some very attractive data measured on a solid state material where Yb4 tetrahedra are isolated within the lattice. This is a very good idea, and there is some beautiful measurements, including field dependent INS data.

Response 3-1: We thank the reviewer for the positive evaluation on our work.

B. The results are interesting, but the authors have missed some significant work in the past that should be discussed and they should read and consider before submitting a revision. Very specifically: there is beautiful work from the Güdel group on use of INS to study lanthanide dimers, including Yb2 dimers. Relevant papers include: Inorganic Chemistry, 1990, 29, 4081; Inorg. Chem., 1994, 33, 1133; PRL, 1989, 62, 210. There is also a recent paper in Nature Communications, 2014, 5, 5243 which discusses exchange interactions in a dysprosium dimer, and which modelled data in a very different way to that proposed here.

Response 3-2: As pointed out by the reviewer, we agree that the previous works on the lanthanide based molecular magnets are quite important in this field. Thus we implemented the description of the significant works on lanthanide based molecular magnets with citation in the introduction and references in the revised manuscript.

C. The data are very good I think. I am far less convinced about the Hamiltonian used and the way data are modelled. Specifically: treating this exchange with an isotropic exchange interaction and then introducing a DM interaction to obtain anisotropy seems odd. I think the authors should have used an anisotropic exchange interaction from the start (as in the Nature Comm paper referenced above). At the very least the authors should try this approach and show that it does not work or is no better.

Response 3-3: We should say that we originally applied the generalized Hamiltonian \mathcal{H}_{gen} of a general 3×3 exchange coupling tensor $J^{\mu\nu}$ for analysis of the obtained INS results. As explained in Supplementary I, the theoretical approach was developed from \mathcal{H}_{gen} . Under the tetrahedral symmetry, the diagonal terms, which correspond to the directional Heisenberg exchange \mathbf{J} -vector (three elements), are represented by two independent J_1 and J_2 values, and the rest six off-diagonal elements are by a J_4 value (DM antisymmetric term) and a J_3 value (pseudo-dipolar symmetric term). Based on \mathcal{H}_{gen} , we could well reproduce all the INS spectra. The best simulations were obtained with $J_2/J_1 = 0.987$ (2), showing that the system has nearly isotropic Heisenberg exchange J , and with almost negligible pseudo-dipolar term ($J_3/J_1 = 0.013$). As the result, the exchange Hamiltonian is effectively reduced to the effective Hamiltonian consisting of an isotropic Heisenberg and a DM interaction term. On the other hand, we also simulated magnetization $M(H)$ and zero-field constant- Q cut $I(\omega)$ by an anisotropic spin exchange XXZ model with fitting parameters of J_1 , J_2 , g_{\parallel} , and g_{\perp} values as the reviewer suggested, but the model does not reproduce the data as shown in the figure below.

We are sorry that this point is not clearly described in the previous manuscript. We modified the manuscript to clarify this point in the main text: Considering the pseudo-spin state of the system, the generalized exchanged Hamiltonian was originally adopted for the analysis of the experimental results. In the analysis, the INS data requires only a few none-trivial non-trivial \mathbf{J} -tensor elements, and that the general Hamiltonian is effectively reduced to the simplified effective Hamiltonian in this system.

Figure. (Left) magnetizations (blue and red lines) $M(H)$ and (right) a zero-field constant-Q cut $I(\omega)$ (red circles) from Fig 1a and Fig 2d in the manuscript, respectively. Cyan colored lines in both panels are simulated by an anisotropic spin exchange XXZ model.

D. Statistics seem fine.

Response 3-4: Thanks.

E. I would like an explanation of why they used the Hamiltonian they used before considering whether conclusions make sense.

Response 3-5: As responded in the 3-3, we are sorry that the discussion and description about the Hamiltonian may not be clear in the previous manuscript. As mentioned, we modified the manuscript.

F. There are two aspects missing. I think there should be an EPR study especially as the g -values used for the techniques modelled vary. I don't understand at all the explanation of the variation in g -values. My guess is the variation is due to the wrong Hamiltonian being used. Secondly, a CASSCF calculation of the g -values of the Yb sites should be reasonably robust in this case. I think some type of electronic structure calculation would add to the paper significantly.

Response 3-6: The g -factors were experimentally estimated from the macroscopic averaged magnetization ($g_{\parallel} = 3.0(1)$ and $g_{\perp} = 2.4(1)$) and the inelastic neutron scattering measurements ($g_{\parallel} = 2.62(2)$ and $g_{\perp} = 2.33(2)$). As suggested by the review, we also carried out the EPR measurements, and the g -factors were estimated to be $g_{\parallel} = 2.54$ and $g_{\perp} = 2.13$ from the analysis of the EPR spectrum. The result is now presented in Supplementary VI. The estimated g -factors somewhat vary depending on the experimental methods, but the ratio consistently merge to $g_{\parallel} / g_{\perp} \sim 1.2$ in all three cases, supporting the anisotropic g -factor in the generalized and effective exchange Hamiltonian. We suspect that the small variations in the absolute g -factor values is mainly due to either the estimation errors or possible contributions of small off-diagonal terms in the g -factor tensor, which cannot be properly determined in the polycrystalline sample and were not taken into account in the analyses.

As also mentioned by the reviewer, the g -factors can be estimated from the ab initio CASSCF calculations. Meanwhile, a cluster crystal field calculation, in which the ligand field effects is also taken into account, has been often employed to explain the local electronic structure of Yb^{3+} ion (Cao H. *et al.*, Phys. Rev. Lett. **103**, 056402 (2009), Malkin B Z, J. Phys.:Condes. Matter **22**, 276003 (2010), Ref. 19: Haku, T. *et al.*, J. Phys. Soc. Jpn. **85**, 034721 (2016)). As shown in the Supplementary IV, we conducted the crystal field calculation for a distorted YbO_6 octahedron under the C_{3v} local symmetry,

and obtained g -factors of $g_{\parallel} = 2.87$ and $g_{\perp} = 2.27$ with the ratio $g_{\parallel}/g_{\perp} \sim 1.26$ by fitting the $4f$ level splittings. These g -factors agree to the experimental values within the variation range of the estimation.

Now we modified the related discussions in the main text and the EPR results are included in Supplementary V.

G. The referencing is weak. Many of the papers referenced are irrelevant, but the very large body of work that has appeared on lanthanide single molecules is largely ignored. The Chibotaru work on Dy3 triangles is certainly relevant here. Also, previous field dependent INS studies are not referenced (e.g. PRL, 2007, 98, 167401). This is easily rectified, and an easy mistake to make when moving into a new area.

Response 3-7: We referred the suggested papers in the revised manuscript.

H. The paper is well written and clear. I don't like Figure 3, which is poor and improvements are needed. I just don't see the relationship between Figure 3a and 3b. In Figure 3a, the energy window observed by INS appears to be moving with field; I don't understand why that would be the case. While the 3D-plot looks attractive, it might be easier for a reader if these data were more simply represented against a common energy axis and with the data for each field plotted against the same window. Figure 3b is rather difficult to understand and I don't think the black background helps. It is not even obvious which transitions are observed. Again, given the model calculated is anisotropic, why is it based on an isotropic exchange interaction plus DM, rather than an anisotropic exchange? At the very least the authors need to demonstrate that an anisotropic exchange interaction would give a less good fit. As the INS was measured on a powder, why give the INS calculated for three principle directions?

Response 3-8: As requested by the reviewer, we modified Fig. 3. The field dependent INS spectra are plotted in a fixed energy window in the modified Fig. 3a. Now one can more easily recognize that the peak position evolves with increase of the magnetic field.

In order to explain the magnetic field H dependent evolution of the INS peak positions, which correspond to the excitation energies, we calculated all the $S_{\text{eff}} = 0, 1,$ and 2 states (shown in Fig. 2e at $H = 0$) and the available transition energies satisfying the INS selection rule as a function of $H (\neq 0)$. As recognized in the calculation results along the three principle axes $[0\ 0\ 1], [1\ 1\ 0]$ and $[1\ 1\ 1]$ shown in Fig. 3b, the transition energy of a certain transition was found to barely vary with the field direction relative to the crystal axis so that it is confined within a finite energy range for an arbitrary field direction. As the results, the directional (powder) averaged INS spectra in Fig. 3a exhibit the transition peaks with finite peak widths, which correspond to the confined energy ranges of the corresponding transitions. The observed INS intensities in Fig. 3a were represented by the grey scale contour map in Fig. 3b, and the intensity scale bar is presented in the inset of the figure (the black color means the zero-intensity).

As discussed above, the theoretical approach was started with the generalized exchange Hamiltonian with a general 3×3 \mathbf{J} -tensor, and the anisotropy in the Heisenberg exchange J turns out to be negligible.

We are sorry that the explanation about Fig. 3b in the main text may not be clear. Thus we modified the related discussions in the revised manuscript.

I think the data already reported could form the basis of a paper in Nature Communications, if the

authors did some more experiments (EPR spectroscopy, CASSCF calculations), read and referenced some relevant previous work, and improved Figure 3.

I'm afraid my review reads more negatively than I would like: I stress - the data are very good and really interesting.

Response 3-9: We addressed all the issues raised by the reviewer in the rebuttal letter, and modified the manuscript accordingly. We believe that the manuscript becomes clearer and stronger. Thanks the reviewer for valuable criticisms and recommendations.

REVIEWERS' COMMENTS:

Reviewer #1 (Remarks to the Author):

The authors have indeed addressed all of my concerns. I am satisfied with their responses except with respect to a small wording detail relating to the description of their M vs. H curves:

- the meaning of "shifts of critical fields" is still not clear in the main text. I suggest rewording to "shifts of the critical fields with respect to the Heisenberg model".

- Authors should more clearly state what aspect of the magnetization has a "Slope" (I guess they mean the plateaus themselves? There is obviously a slope at the transitions, even in the heisenberg exchange model - likely due to the powder averaging and anisotropic g-tensor)

Another wording point - I disagree with their response to one of the other referees regarding whether this material is a pyrochlore or not. It is a "breathing pyrochlore", not a pyrochlore in the strict sense. The difference is that the "down" tetrahedra are larger than the "up" tetrahedra. It should not be referred to as a pyrochlore without the "breathing" qualifier.

Finally, the most important point is that I believe the authors are correct in their assertion that their work goes above and beyond the previously published works on the same material (Rau et al [36], Haku et al [37]). This work alone has investigated the effect of a magnetic field on the molecular Seff states. In particular, they have shown that the system has avoided level crossings, and thus may be of interest in the quantum computation world.

Reviewer #2 (Remarks to the Author):

O.k. from my side, can be published.

Reviewer #3 (Remarks to the Author):

The authors have tried to answer the many comments from the three reviewers and I am satisfied that most of these answers are sensible - at least from my perspective.

My only remaining concern are the g-values reported. The authors must state which are the correct g-values, not just that there is a disagreement between them. This will be confusing for a future reader. My very strong preference are for the values measured by EPR spectroscopy which are close to those used for INS. The magnetisation measurement is less reliable for this parameter. I think the authors just need to clarify this and then publication can proceed.

REVIEWERS' COMMENTS:

Reviewer #1 (Remarks to the Author):

The authors have indeed addressed all of my concerns. I am satisfied with their responses except with respect to a small wording detail relating to the description of their M vs. H curves:

- the meaning of "shifts of critical fields" is still not clear in the main text. I suggest rewording to "shifts of the critical fields with respect to the Heisenberg model".

Response 1-1: Thanks for pointing out ambiguity in “shifts of critical fields”. As suggested by the reviewer, we reworded it as “shifts of the critical fields with respect to the Heisenberg model” in the revised manuscript.

- Authors should more clearly state what aspect of the magnetization has a "Slope" (I guess they mean the plateaus themselves? There is obviously a slope at the transitions, even in the heisenberg exchange model - likely due to the powder averaging and anisotropic g-tensor)

Response 1-2: “Slope” means distinctive increment of magnetization in the range of applied magnetic field $0 < H < H_{C1}$. The origin of “Slope” is a singlet-triplet admixed ground state due to the DM interaction and the Zeeman term with anisotropic g-factors which do not commute with the Heisenberg interaction. Although both can induce a finite paramagnetic response to the applied magnetic field in the range $0 < H < H_{C1}$, the DM interaction is more significant than the anisotropic Zeeman term, which is now provided in the revised Supplementary Fig. 5 and Note 2. This point is also clearly stated in the revised manuscript.

Another wording point - I disagree with their response to one of the other referees regarding whether this material is a pyrochlore or not. It is a "breathing pyrochlore", not a pyrochlore in the strict sense. The difference is that the "down" tetrahedra are larger than the "up" tetrahedra. It should not be referred to as a pyrochlore without the "breathing" qualifier.

Response 1-3: We agree that a terminology of “pyrochlore” is used improperly for this system. We replace it by “breathing pyrochlore” in the revised manuscript as the reviewer suggested.

Finally, the most important point is that I believe the authors are correct in their assertion that their work goes above and beyond the previously published works on the same material (Rau et al [36], Haku et al [37]). This work alone has investigated the effect of a magnetic field on the molecular Seff states. In particular, they have shown that the system has avoided level crossings, and thus may be of interest in the quantum computation world.

Response 1-4: We addressed all the issues raised by the reviewer in the rebuttal letter, and modified the manuscript accordingly. Thanks again the reviewer for valuable criticisms and recommendations.

Reviewer #2 (Remarks to the Author):

O.k. from my side, can be published.

Response 2-1: We appreciate the reviewer's approval to publish the paper.

Reviewer #3 (Remarks to the Author):

The authors have tried to answer the many comments from the three reviewers and I am satisfied that most of these answers are sensible - at least from my perspective.

Response 3-1: We thank the reviewer's positive response to the previous revision.

My only remaining concern are the g-values reported. The authors must state which are the correct g-values, not just that there is a disagreement between them. This will be confusing for a future reader. My very strong preference are for the values measured by EPR spectroscopy which are close to those used for INS. The magnetisation measurement is less reliable for this parameter. I think the authors just need to clarify this and then publication can proceed.

Response 3-2: We agree that g-values obtained by the EPR spectroscopy are the most reliable among experimentally obtained ones. The clear statement for the proper g-values is now added in the revised manuscript.